# Resistance Mechanisms of Fluoroquinolone in *Escherichia coli* Isolated from Taihe Black-Boned Silky Fowl Exhibiting Abnormally Slow Fluoroquinolone Metabolism in Jiangxi, China

**DOI:** 10.3390/antibiotics14090955

**Published:** 2025-09-21

**Authors:** Li Zhang, Mengjun Ye, Yifan Dong, Lijuan Yuan, Jianjun Xiang, Xiren Yu, Qiegen Liao, Qiushuang Ai, Suyan Qiu, Dawen Zhang

**Affiliations:** 1Institute of Quality & Safety and Standards of Agricultural Products Research, Jiangxi Academy of Agricultural Sciences, Nanchang 330200, China; zhangli9061@163.com (L.Z.); 13668043636@163.com (M.Y.); dongyifan0321@163.com (Y.D.); ylj198820062467@163.com (L.Y.); yxjj101@sina.com (J.X.); nkyyxr7012@163.com (X.Y.); liaoqg2004@126.com (Q.L.); qai12539@163.com (Q.A.); qiusuyan@126.com (S.Q.); 2Key Laboratory for Quality and Safety Control of Poultry Products, Ministry of Agricultural Products Research, Jiangxi Academy of Agricultural Sciences, Nanchang 330200, China; 3Jiangxi Provincial Key Laboratory for Quality and Safety of Agricultural Products, Nanchang 330200, China

**Keywords:** Taihe Black-Boned Silky Fowl, *Escherichia coli*, fluorquinolone resistance mechanism

## Abstract

Objectives: The Taihe Black-Boned Silky Fowl (TBSF) is a unique indigenous chicken breed in China, characterized by widespread melanin deposition throughout its body. Fluoroquinolones (FQs) such as enrofloxacin can persist in TBSF for an extended period exceeding 100 days. The aim of this study was to examine the current status and development trends of FQ resistance within the TBSF breeding environment. Methods: Whole-genome sequencing was utilized to identify the molecular presence of quinolone resistance-determining region (QRDR) mutations and plasmid-mediated quinolone resistance (PMQR) genes in *Escherichia coli* isolates obtained from TBSF farms. Network inference based on strong Spearman correlations (ρ > 0.5) and statistically significant associations (*p*-value < 0.05) was applied to investigate the co-occurrence patterns among FQ residues, resistance phenotypes, and antibiotic resistance genes. Results: The results showed that FQ residues were identified as the primary contributor to FQ resistance in *E. coli* isolates. Mutations at QRDR sites were the predominant factor driving FQ resistance, rather than PMQR determinants. This study also reported the first identification of GyrA-S83Q mutation being associated with FQ resistance. Conclusions: It was concluded that *E. coli* strains in TBSF environments, where chickens have a long-term residual metabolic cycle of antimicrobials, may develop and evolve new mechanisms to adapt to this environment. Further research is warranted to investigate the evolution of FQ resistance in *E. coli* strains within TBSF environments.

## 1. Introduction

Antimicrobial resistance (AMR) constitutes a major global health challenge [1,2]. The inappropriate and excessive use of antibiotics is among the key drivers contributing to this escalating issue [3]. Globally, approximately 73% of all antimicrobials sold are utilized in food-producing animals [4]. The poultry industry represents one of the largest sectors in terms of antibiotic consumption [5]. Fluoroquinolones (FQs), characterized by their broad-spectrum antimicrobial activity and excellent bioavailability following administration, demonstrate high efficacy against Gram-negative bacteria such as *E. coli* and *Salmonella* spp. [6]. Consequently, they have been extensively employed in veterinary medicine within the animal production sector [7]. As a result, FQ-resistant bacteria are frequently detected in poultry environments and poultry meat products [8,9]. The prevalence of FQ resistance is notably higher in poultry environments compared to other food-animal production settings [9,10,11].

*E. coli* serves both as a common commensal organism and as a pathogen in animals and humans [12]. As the leading pathogen associated with resistant-related deaths, *E. coli* presents a substantial public health concern [13]. Among FQ-resistant Gram-negative bacteria, *E. coli* has consistently been identified as the predominant isolate [14]. It constitutes a major reservoir of antibiotic resistance genes (ARGs) and plays a critical role in the horizontal transfer of ARGs and virulence factor genes (VFGs) between zoonotic and other bacterial species [15]. The degree of resistance in *E. coli* is regarded as a reliable indicator for assessing resistance dynamics in various environments. Investigating the mechanisms underlying FQ resistance in *E. coli* across diverse ecological contexts is of paramount importance and can facilitate the development of effective strategies to mitigate antimicrobial resistance [16,17].

FQ resistance evolves rapidly in the resident coliform flora of animals during treatment with FQs [18]. Antibiotic administration confers selective advantages, leading to significantly elevated expression levels of efflux pump genes in *E. coli* isolates from chicken feces [19]. Subtherapeutic concentrations of antimicrobials are strongly suspected to contribute to the development of antimicrobial resistance [20]. Environmentally relevant concentrations of antibiotics can rapidly induce heritable antibiotic resistance. De novo evolved mutants were capable of acquiring FQ resistance within 10 days when exposed to subtherapeutic concentrations of enrofloxacin (ENR) and ciprofloxacin (CIP) [21].

Taihe Black-Boned Silky Fowl (TBSF) is a distinctive local chicken breed in China, characterized by extensive melanin deposition in the skin, meat, and bones. In our previous studies, we demonstrated that certain antibiotics, particularly FQs such as ENR, can persist in TBSF for an extended period exceeding 100 days [22]. Unlike broiler chickens, the average rearing period for TBSF on farms is approximately 120 days before slaughter, with some batches even being raised for up to three to five years or even longer. Studies have shown that long-term antibiotic treatment is a major driver of target mutations in the gastrointestinal tract [23,24]. Even very low doses of ENR (1/1000 therapeutic dose) can also select for mutations during the later stages of treatment [24]. Under specific conditions or in response to the selective pressure of novel antimicrobial agents, the emergence of novel mechanisms of FQ resistance may occur [23,25]. However, limited information is available regarding FQ-resistant *E. coli* from TSBF, where the mechanisms of FQ resistance may differ from those observed in *E. coli* isolated from broiler chickens and other environments. Further research is warranted to investigate the mechanisms of FQ resistance in *E. coli* within the TBSF environment, where FQ metabolism occurs at an extremely slow rate.

The primary objective of this study was to investigate the current status and development of FQs resistance in TBSF environments characterized by a distinct and slow metabolic cycle of FQs. Whole-genome sequencing (WGS) was employed to detect the molecular presence of quinolone resistance-determining region (QRDR) mutants and plasmid-mediated quinolone resistance (PMQR) genes in *E. coli* strains isolated from TBSF farms, thereby elucidating the molecular mechanisms of FQ resistance and their associated influencing factors.

## 2. Results

### 2.1. E. coli FQ-Resistant Phenotypes

A collection of 34 *E. coli* strains was isolated from TBSF farms. These isolates were isolated from feces (n = 20), soil (n = 10), and feed (n = 4). More than half of the strains (52.9%, 18 out of 34) demonstrated reduced susceptibility to at least one FQ (Table 1 and Appendix A). Specifically, 52.9% strains showed decreased susceptibility to flumequine (UB), 41.1% to moxifloxacin (MXF), 17.6% to ENR, 8.8% to CIP, and 5.9% to norfloxacin (NOR) and levofloxacin (LVX). Notably, two *E. coli* strains isolated from fecal samples exhibited resistance to all six FQs tested. Among these strains, twelve were isolated from fecal samples, five from soil samples, and one from a feed sample.

### 2.2. Prevalence and Genetic Environment of PMQR Determinants Among E. coli Strains

PMQR determinants were exclusively identified in strains exhibiting reduced susceptibility to FQs. However, not all FQ-susceptible strains harbored PMQR determinants. Only nine strains possessed the FQ resistance gene *qnr*S1, of which five were isolated from fecal samples, three from soil samples, and one from a feed sample. No additional PMQR determinants were detected in these *E. coli* strains. Notably, neither of the two *E. coli* strains resistant to all six FQs carried any PMQR determinants (Table 2). The location of *qnr*S1 genes was investigated using whole-genome sequencing (WGS). The most prevalent type of transferable plasmids carrying *qnr*S1 was the pFS13Z2S plasmid, which belongs to the IncHI2 incompatibility group. In *E. coli* strains, *qnr*S1 genes were consistently flanked by various mobile genetic elements, including transposase genes, insertion sequence IS1, resolvase genes, and recombinase genes. Genetic mapping revealed that an IS2 transposase gene was invariably located upstream of *qnr*S1, while a Tn552 transposon resolvase gene was consistently positioned downstream (Appendix A).

### 2.3. Analysis of Mutations in the QRDRs

The DNA sequences of the QRDRs were analyzed in *E. coli* isolates from TBSF farms to identify point mutations associated with amino acid substitutions (Figure 1 and Appendix A). Chromosomal mutation sites within the QRDRs were detected not only in FQ-resistant strains but also in some FQ-susceptible strains, with varying mutation patterns observed. Two *E. coli* isolates resistant to six FQs exhibited three mutations in DNA gyrase and topoisomerase IV: one in GyrA at positions 83 (Ser → Leu) and 87 (Asp → Asn), and another in ParC at position 80 (Ser → Ile). A strain resistant to CIP, ENR, MXF and UB exhibited a single mutation in DNA gyrase GyrA at position 83 (Ser → Glu). In contrast, a strain resistant solely to MXF and UB exhibited a single mutation in DNA gyrase GyrA at position 83 (Ser → Leu). Besides the mutations at positions 83 and 87 in GyrA and position 80 in ParC, additional mutations were identified at positions 422 and 535 in GyrA and positions 406 and 437 in ParC in some *E. coli* isolates.

### 2.4. FQ Residues in TBSF Samples

The residues of three FQs in TBSF samples were analyzed to assess antibiotic usage on *E. coli*—positive TBSF farms (Appendix A). In four *E. coli*—positive TBSF farms, FQ residues were detected. FQ-nonsusceptible *E. coli* strains were identified in all of these farms. A strain of *E. coli* isolated from a fecal sample exhibited resistance to all tested FQs in the antimicrobial susceptibility assay. Furthermore, ENR residues were detected in the same fecal sample, as well as in chicken breast and egg samples collected from the corresponding farm, at concentrations of 7.29 μg/kg, 10.48 μg/kg, and 1.04 μg/kg, respectively. In a second farm, another *E.coli* strain isolated from a fecal sample showed resistance to all six tested FQs, while both ENR and CIP were detected in the breast meat samples. Ofloxacin (OFX) was detected in both fecal and breast meat samples from one TSFB farm, where an *E. coli* strain isolated from a soil sample exhibited resistance to ENR, MXF, and UB.

### 2.5. Correlation Between FQ Residues, Resistance Phenotypes, and ARGs

The co-occurrence patterns among FQ residues, resistance phenotypes, and antibiotic resistance genes (Appendix A) were analyzed using network inference based on strong Spearman correlations (ρ > 0.5) and statistically significant associations (*p*-value < 0.05) (Figure 2). The most densely connected node in the network indicated a significant association between ENR residues in TBSF farms and their corresponding resistance genes and phenotypes. Mutations at positions S83L and D87N in GyrA and S80I in ParC were found to contribute more substantially to FQ resistance than the mobile PMQR gene *qnr*S1 and other mutation sites within DNA gyrase and topoisomerase IV. In this study, the antibiotic efflux pump multidrug resistance protein MdtH showed no correlation with FQ resistance phenotypes. Additionally, the S83Q mutation in GyrA, potentially influenced by the presence of the beta-lactamase OXA-10, demonstrated a significant association with CIP resistance.

## 3. Discussion

### 3.1. The Prevalence of FQ-Resistant E. coli Strains Across Different Environments

In this study, 52.9% of *E. coli* isolates obtained from TBSF farms were nonsusceptible to FQs, a rate that was notably lower than the FQ-resistance levels observed in other broiler farms across China [26,27]. One possible explanation for this difference was the more frequent use of FQs in conventional intensive broiler farming systems. In contrast, TBSF primarily employed free-range rearing methods in forested environments, combined with the prohibition of FQs use in TBSF production in China, which may contribute to reduced FQs usage and a lower prevalence of antimicrobial resistance. The FQ-resistance rate of *E. coli* observed in this study was significantly lower than that isolated from symptomatic birds, which may be attributed to the more frequent use of FQs in the treatment of bacterial infection in the latter group [28].

The FQ-resistance rates of *E. coli* isolates from various poultry sources and geographic regions to FQs exhibited significant variability. The prevalence of ciprofloxacin-resistant *E. coli* in broiler chicken and broiler farm environments, based on samples collected from 30 broiler farms across 18 upazilas (administrative sub-districts) in Chattogram, Bangladesh, was 77.6% and 88.8%, respectively [29]. 6.8% (76/1122) *E. coli* isolates from broiler carcasses collected in slaughterhouses in different geographical areas in Canada were resistant to FQs [30]. 52.9% of *E. coli* isolates from chickens were resistant to ENR from Hungarian poultry farms between 2022 and 2023 [31]. The *E. coli* isolates obtained from 181 caecal samples of slaughtered broilers in Ilorin, Kwara State, Nigeria, exhibited complete (100%) resistance to CIP [32]. In contrast, the majority (98%) of the *E. coli* isolates from in table eggs sold at retail supermarkets in Western Australia were susceptible to CIP [33].

Although the detection rates vary from place to place, numerous studies have demonstrated a significant increase in the prevalence of FQ-resistant bacteria during poultry production, where FQs were frequently employed as primary agents for prevention and treatment of bacterial infections [4,26,34]. Given the ongoing use of FQs as therapeutic or prophylactic agents in many countries, the prevalence of FQ resistance among *E. coli* isolates from poultry remains significantly elevated [29,35].

### 3.2. Epidemiological Characteristics of PMQR Genes

The *qnrS* gene was consistently identified as the predominant PMQR gene among *E. coli* isolates obtained from poultry-related environments [36]. Additionally, *oqx*A/B and *aac*(6′)-*Ib-cr* were frequently detected in samples derived from poultry sources [37]. In the present study, only *qnr*S1 genes were detected in *E. coli* isolates from TBSF farms, with no other PMQR genes identified. Furthermore, the *qnr*S1 genes were found to be flanked by diverse mobile genetic elements, suggesting their potential for horizontal gene transfer mediated by these elements.

In the present study, *E. coli* strains lacking mutations in the QRDR exhibited low-level resistance to FQ antimicrobials, even when carrying PMQR determinants. This observation aligns with previous studies, which have demonstrated that PMQR genes, including *qnr*, *aac*(6′)-*Ib-cr*, *qep*A, and *oqx*AB, confer only low-level resistance to FQs [16,38]. However, other prior investigations have shown that the widespread prevalence of PMQR determinants has served as a driving force in the selection for high-level quinolone resistance [39]. These PMQR determinants often facilitated the emergence of elevated resistance levels by promoting mutations within the QRDR [40]. The presence of PMQR genes might enhance the mutation frequency of QRDR in *E. coli*, thereby contributing to the development and progression of FQ resistance [41].

Previous studies have demonstrated that bacterial strains carrying PMQR genes in combination with mutations in QRDR determinants typically exhibit high level FQ resistance [39,42]. However, in the present study, the two strains displaying the highest level of FQ resistance did not carry any PMQR genes. These strains exhibited only specific mutations in GyrA (codons 83 and 87) and ParC (codon 80). This observation may suggest that, as FQs continue to evolve and are extensively used in poultry production, the resistance mechanisms of *E. coli* to FQs have undergone subtle shifts within the poultry environment. The resistance pattern appeared to have transitioned from being primarily driven by PMQR-mediated horizontal gene transfer to being predominantly governed by chromosomal mutations in QRDR determinants. This evolutionary shift was primarily driven by the fitness cost associated with maintaining resistance traits in bacterial populations [43]. Various FQ-selective pressures significantly affected the fitness of alternative mutations and determined the sequential order of mutation events [44]. In TBSF environments, considering the prolonged withdrawal period of FQs, it is hypothesized that PMQR determinants initially facilitate mutations within the QRDR of *E. coli*. Subsequently, during the long-term evolutionary adaptation of these isolates under sustained FQ selective pressure, QRDR mutations were selectively favored due to their lower fitness costs. This resistance acquisition mechanism reduced the survival burden imposed by PMQR and enhanced the competitive fitness of resistant strains over susceptible counterparts for ecological niches within the host [45].

### 3.3. Mutations in QRDR Determinants of E. coli Strains

The resistance-associated mutator strains demonstrated a selective advantage even under low antibiotic concentrations [46]. The mutant selection window for these strains was broader compared to that of wild-type strains, extending toward both higher and lower antibiotic concentrations. At concentrations that did not inhibit bacterial growth, mutations in GyrA and GyrB occurred at frequencies comparable to those observed at inhibitory concentrations. Mutations in ParC typically arise subsequent to GyrA mutations, potentially compensating for the fitness cost associated with antimicrobial resistance [46,47]. According to prior studies, the emergence of high-level resistance to FQs typically involved multiple mutational events [39,48]. Mutations in the *gyr*A and *par*C genes within the QRDR represented the primary molecular mechanism underlying FQ resistance in *E. coli*. The most commonly reported mutation sites *E. coli* strains were S83L and D87N in GyrA, and S80I in ParC [48]. In the present study, the two *E. coli* isolates did not carry any PMQR determinants but possessed three specific mutations—S83L and D87N in GyrA and S80I in ParC—and exhibited resistance to all tested FQs. This observation aligns with findings from previous investigations, which have demonstrated that FQ-resistant *E. coli* isolates harboring dual *gyr*A mutations (at codons 83 and 87) in conjunction with *par*C mutations display markedly elevated minimum inhibitory concentration (MIC) values [2,49].

Previous studies have shown that single mutations in QRDR determinants confer only a minimal impact on FQs resistance level [39,44,50]. Even in the presence of both the *qnr*S1 gene and the amino acid substitution S83L in the *gyr*A gene, *E. coli* strains exhibited resistance only to nalidixic acid, but not to ENR or other FQs [51]. In our study, one *E. coli* isolate carrying only the GyrA-S83L mutation and lacking PMQR genes exhibited resistance exclusively to UB. This observation was consistent with previously published findings [50,51]. However, another *E. coli* isolate possessing PMQR genes and harboring only the GyrA-S83Q mutation demonstrated markedly increased resistance to CIP, MXF and UB, while showing decreased susceptibility to ENR. These results suggest that a single mutation may also contribute to elevated resistance levels against FQs, a finding that is not entirely in agreement with prior literature.

To the best of our knowledge, GyrA-S83Q represents a novel mutation associated with FQ resistance and has not been previously reported in the literature. As demonstrated by prior studies, de novo mutants can be selected under sub-MIC levels of antibiotics [21,25,47]. Subtherapeutic doses of FQs might accelerate the development of antimicrobial resistance by promoting bacterial evolution [19]. The emergence of this novel GyrA-S83Q mutation may be associated with the prolonged withdrawal period of FQs in TBSF environments.

These compensatory mutations might contribute to the stabilization of resistance within bacterial populations, as resistance-associated mutations did not necessarily impose significant fitness costs on bacterial growth [52]. Enhanced resistance to FQs could continue to emerge even in the absence of further antimicrobial exposure [52,53]. Therefore, future research should focus on evaluating FQ resistance in *E. coli* isolates obtained from TBSF environments, with particular emphasis on the role of the novel GyrA-S83Q mutation and the potential synergistic interactions between single QRDR mutations and PMQR determinants.

### 3.4. Factors Influencing FQ Resistance in E. coli Strains

Numerous factors might influence the FQ resistance of *E. coli* strains. These factors not only directly affected FQ resistance but were often intercorrelated [54]. In this study, based on the results of network analysis, the level of FQ resistance was found to be strongly associated with ENR residues in TBSF farms. Many previous studies have also demonstrated that the administration of FQs could readily induce FQ resistance in *E. coli* isolates [18,19]. A reduction in FQs usage has been significantly correlated with a decrease in the prevalence of FQ-resistant *E. coli* isolates [55,56].

The co-selection mechanism plays a more prominent role in promoting antimicrobial resistance associated with overall antibiotic use compared to the selective pressure exerted by individual antibiotic classes [55,57]. The use of one class of antimicrobials might lead to the selection of resistance to another unrelated class, a phenomenon referred to as co-resistance. This explained why most FQ-resistant *E. coli* isolates from poultry farms exhibited multidrug resistance. The cross-resistance of FQ-resistant *E. coli* strains to beta-lactams, florfenicol, tetracycline, and other antibiotics has raised considerable public health concerns [46,58]. ESBL-producing isolates were commonly found to carry a combination of the PMQR genes *qnr* and *aac*(6′)-1*b-cr* [36,58,59]. Higher prevalence rates of PMQR genes, particularly *qnr*, have been observed in ESBL-positive *E. coli* strains. In this study, we found that the beta-lactamase OXA-10 might contribute to the S83Q mutation in GyrA, which leads to CIP resistance in *E. coli* strains.

Factors that influence the activity of efflux pumps or induce alterations in outer membrane proteins might also contribute to FQ resistance in *E. coli* strains. Overexpression of efflux pumps in *Enterobacteriaceae* represented a major mechanism underlying bacterial resistance to multiple antimicrobial agents. Several of these efflux systems have been implicated in FQ resistance [54]. In our study, although the antibiotic efflux pump multidrug resistance protein MdtH showed no association with FQ resistance phenotypes in network analysis, the functional role of antibiotic efflux pumps cannot be overlooked. This is particularly relevant given that many multidrug *E. coli* isolates possess the mutations of *mdt*H, including the two strains that exhibited resistance to all tested FQs.

Farming practices and management strategies implemented on poultry farms have played a significant role in the emergence of FQ-resistant bacteria. A higher prevalence of resistance to FQs and multidrug resistance has been observed in conventional farms compared to organic farms [60,61]. Poor hygiene practices and inadequate biosecurity measures may contribute critically to the persistence of FQ-resistant *E. coli* populations. It is possible that disinfectants also facilitate the dissemination of FQ resistance in *E. coli* [32]. Seasonal variation may also influence the FQ resistance rate in *E. coli*, with the probability of resistance development being approximately twice as high in summer compared to winter. This trend could be associated with the increased incidence of infectious diseases and higher antibiotic usage during warmer months [62].

In addition to the aforementioned factors, a multitude of additional elements have contributed to the emergence and development of FQ resistance. Different *E. coli* serotypes tend to favor specific mutation sites to acquire resistance to FQs [30]. Transnational transmission, as well as temporal and regional variations, can also influence the prevalence and distribution of FQ-resistant *E. coli* strains [63]. Vertical transmission of FQ-resistant strains commonly occurs within poultry farms and across the broader poultry production chain [2,64]. Therefore, this highlighted the need for comprehensive strategies that go beyond the regulation of antibiotic use in order to effectively mitigate the spread of FQ-resistant bacteria, such as comprehensive disinfection protocols [65].

## 4. Materials and Methods

### 4.1. Sampling, Bacterial Isolation, and FQs Detection

Fecal, soil, feed, breast meat, and egg samples were collected in November 2022 from 18 healthy TBSF farms situated in Taihe, Jiangxi Province, China. Egg samples were collected in TBSF laying farms. The breast meat, feed, and egg samples were collected from the same batch as the fecal samples. The background information of the farms was provided in Figure 3 and Appendix A. *E. coli* was isolated from fecal, soil and feed samples. Samples were collected aseptically using sterilized utensils and placed into sterilized plastic tubes or bags, then transported to the laboratory within 24 h in ice-cooled containers. Immediately upon arrival at the laboratory, the isolation of *E. coli* was initiated. Ten grams of fecal sample were added to a sterile conical flask containing 90 mL of sterilized nutrient broth and 25 g of soil or feed sample were added to a sterile plastic bag containing 225 mL of sterilized nutrient broth. The mixture was thoroughly shaken and well mixed or homogenized to ensure complete homogenization. The samples were then incubated for 6 h at 36 °C. Following incubation, 10 μL of the culture was transferred into a sterile conical flask containing 30 mL of intestinal bacterial enrichment broth and incubated at 42 °C ± 1 °C for 18 h. Subsequently, one loopful of the culture was streaked onto eosin methylene blue agar and incubated for 24 h at 37 °C. After confirmation of suspicious colonies via biochemical testing, the isolates were submitted for whole-genome sequencing.

As the use of antibiotics, particularly through feed administration, represents one of the primary driving factors for the emergence and development of drug resistance, residual antibiotics in feces exert selective pressure on the intestinal microbial community, thereby promoting the proliferation of drug-resistant strains. Residues of FQs, including ENR, CIP, and OFX, in fecal, breast meat, egg, and feed samples were analyzed using the methodology described in our previous study, as described in the Appendix A [22]. The instrumental analysis was conducted using an LC-30AD ultra-performance liquid chromatography (Shimadzu, Kyoto, Japan), coupled to a QTRAP 6500 triple quadrupole mass spectrometer (SCIEX, Framingham, MA, USA).

### 4.2. FQs Susceptibility Testing

All *E. coli* isolates were subjected to antimicrobial susceptibility testing against six FQs. The Kirby-Bauer disc diffusion method was employed to determine the inhibition zone diameters, and the results were interpreted in accordance with the Clinical and Laboratory Standards Institute document M100 [66]. The reference strain *E. coli* ATCC25922 was used as a quality control. The antibiotic test panel comprised the following disks with their respective concentrations: ENR (10 μg), CIP (5 μg), UB (30 μg), NOR (10 μg), MXF (5 μg), and LVX (5 μg).

### 4.3. Whole-Genome Sequencing and Identification of ARGs

The whole-genome scans of all *E. coli* strains were obtained through de novo sequencing of the bacterial genome, which was performed at Majorbio Bio-Pharm Technology Co., Ltd. (Shanghai, China) using the advanced Illumina novaseq X plus platform. We performed de novo assembly using SOAPdenovo v2.04 [67]. To further enhance the quality of the assembled genome, GapCloser v1.12 was employed to close gaps that appeared during the scaffolding process. This step made use of the abundant paired-end relationships of short reads, ultimately improving the completeness of the genome sequences. The coding sequences of the genome (CDS) were predicted using the Glimmer 3.02 [68], GeneMarkS V4.3 (http://topaz.gatech.edu/GeneMark, accessed on 13 May 2024), and Prodigal V2.6.3 (https://github.com/hyattpd/Prodigal, accessed on 13 May 2024) software. The nucleic acid and amino acid sequences of functional genes were subsequently obtained through these predictions. All *E. coli* isolates were screened for the presence of plasmid-mediated antibiotic resistance determinants, including quinolone, β-lactam, tetracycline, chloramphenicol, aminoglycoside, sulfonamide, and multi-drug resistance genes, as previously described [27]. Specifically, the identification of these genes from DNA sequences was conducted using DIAMOND v0.8.35, based on curated data obtained from the Comprehensive Antibiotic Resistance Database (version 3.2.6). Resistance determinants were identified using stringent criteria requiring a minimum of 80% amino acid identity, at least 50% sequence length identity, and a coverage of no less than 50% relative to known resistance proteins. The QRDRs of *gyr*A/*gyr*B and *par*C/*par*E genes were analyzed for mutations in these isolates. WGS data for all *E. coli* isolates used in this study were submitted to the National Center for Biotechnology Information (NCBI) under BioProject accession number PRJNA1135707. Accession numbers for individual isolates were provided in Appendix A.

### 4.4. Correlation Between FQ Residues, Resistance Phenotypes and ARGs

Numerical values were assigned to FQ resistance, ARGs, and their corresponding phenotypes, with a score of 2 designated for resistant phenotypes, 1 for positive items or intermediate phenotypes, and 0 for negative or susceptible phenotypes. The presence of known resistance genes corresponding to FQ resistance was considered indicative of positive ARGs in the *E. coli* isolates. Conversely, the absence of such ARGs was considered indicative of a negative result. If FQ residues were detected in any sample originating from a specific farm, the *E. coli* strain isolated from any sample of that farm would be classified as drug residue positive. Conversely, if no FQ residues were detected, the *E. coli* strain would be classified as drug residue negative. The data were analyzed using the online tools provided by the Majorbio Cloud Platform (https://cloud.majorbio.com/page/tools/, accessed on 14 April 2025).

## 5. Conclusions

This study investigated FQ-resistant *E. coli* isolates obtained from TBSF farms. FQ-resistant *E. coli* isolates were prevalent in TBSF farms and may persist as contaminants in these environments for years following the cessation of on-farm FQ use. It was noteworthy that LVX and NOR have been explicitly banned in China for use in food-producing animals. However, two *E. coli* isolates obtained from TBSF fecal samples in this study exhibited resistance to these prohibited FQs. FQ residues were identified as the primary contributor to FQ resistance in *E. coli* isolates. Mutations at QRDR sites were the predominant factor driving FQ resistance, rather than PMQR determinants. This study also reported the first identification of the GyrA-S83Q mutation being associated with FQ resistance. These findings suggested that *E. coli* strains present in environments where chickens undergo a long-term residual metabolic cycle of antimicrobials might develop and evolve novel mechanisms to adapt to such conditions and to emerging antimicrobial agents. However, how the related ARGs and FQ residues induce the development and evolution of novel resistance mechanisms remains unclear in this study. Further research is warranted to investigate the evolution dynamics of FQ resistance in *E. coli* isolates under prolonged exposure to FQ or other antibiotics within TBSF environments.

## Figures and Tables

**Figure 1 antibiotics-14-00955-f001:**
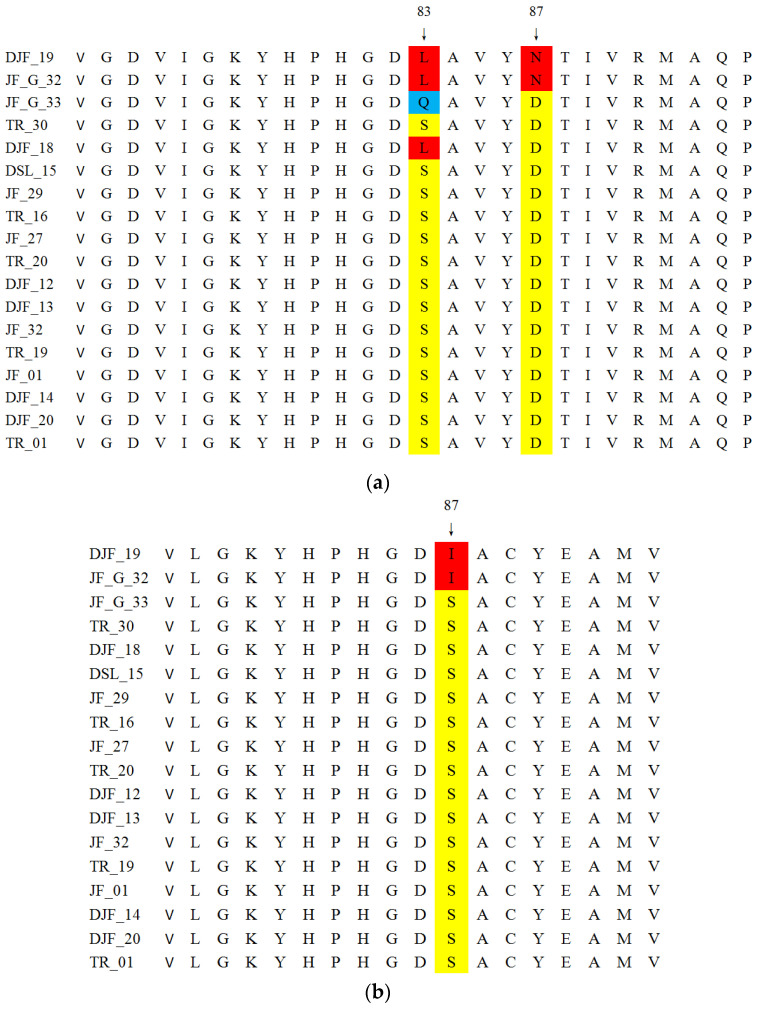
Multiple sequence alignment of GyrA (**a**) and ParC (**b**) across *E.coli* isolates from TBSF farms. Non-mutant amino acid sites were highlighted in yellow, whereas mutations were indicated in red and blue.

**Figure 2 antibiotics-14-00955-f002:**
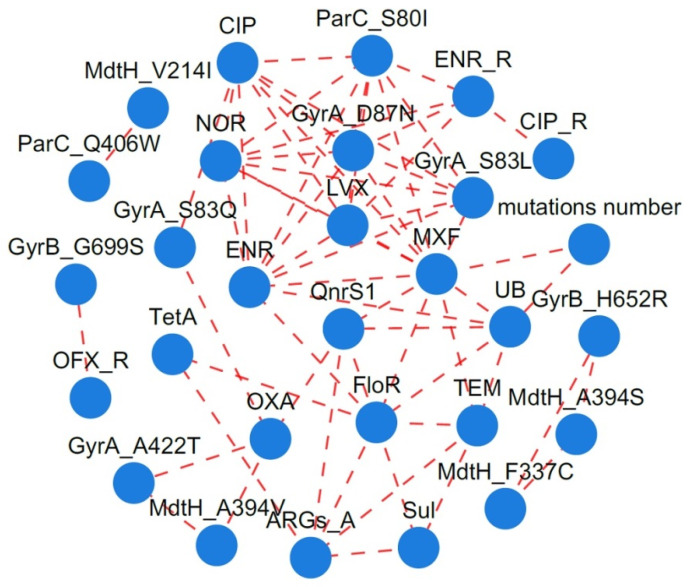
The network analysis revealing the co-occurrence patterns among FQ residues, ARGs, and mutations in the QRDRs (Spearman coefficient, ρ > 0.5, *p* < 0.05). CIP/NOR/LVX/ENR/MXF: FQ-resistant phenotypes; ENR_R/CIP_R/OFX_R: drug residue positive; QnrS1: plasmid-mediated quinolone resistance protein; FloR: chloramphenicol exporter protein; TEM: broad-spectrum beta-lactamase; OXA: beta-lactamase; ARGs_A: aminoglycoside acetyltransferase/aminoglycoside acetyltransferase; Sul: sulfonamide-resistant dihydropteroate synthase; others: mutation points.

**Figure 3 antibiotics-14-00955-f003:**
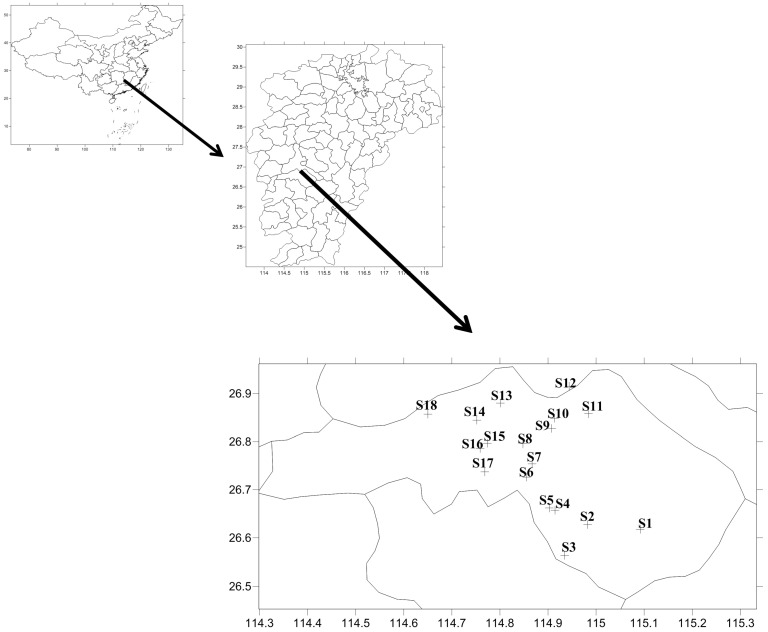
Geographical locations of TBSF farms.

**Table 1 antibiotics-14-00955-t001:** FQ Resistance Phenotype of *E. coli* Isolates.

Resistance Phenotype	NOR	ENR	CIP	LVX	MXF	UB	Number of Strains	Strain Names
1	R	R	R	R	R	R	2	DJF_19/JF_G_32
2	S	I	R	S	R	R	1	JF_G_33
3	S	I	S	S	I	R	2	TR_30/DJF_18
4	S	S	S	S	I	R	1	DSL_15
5	S	I	S	S	I	I	1	JF_29
6	S	S	S	S	I	I	6	TR_16/TR_20/JF_27/DJF_13/DJF_20/TR_01
7	S	S	S	S	S	I	5	DJF_12/JF_32/TR_19/JF_01/DJF_14
8	S	S	S	S	I	S	1	JF_33
9	S	S	S	S	S	S	15	others

“S” indicates susceptibility. “I” indicates intermediate susceptibility. “R” indicates resistance.

**Table 2 antibiotics-14-00955-t002:** PMQR Determinants among *E. coli* Isolates.

Stain No.	PMQR Determinants	Location Scaffold	Gene No.	Plasmid Type	Acc No.	Acc Plasmid Name
JF_G_33	*qnr*S1	44	4388	IncHI2	NZ_KY421937.1	pFS13Z2S
TR_16	*qnr*S1	54	4820	IncHI2	NZ_KY421937.1	pFS13Z2S
JF_27	*qnr*S1	90	4535	IncHI2	NZ_KY421937.1	pFS13Z2S
DJF_12	*qnr*S1	99	4603	IncX1	NZ_CP037995.1	psg_ww281
JF_29	*qnr*S1	34	3995	IncX1	NZ_CP022964.1	pQJDsal1
TR_30	*qnr*S1	29	4575	IncFIB(K)	NZ_LR213454.1	3
TR_20	*qnr*S1	68	4415	IncFIC(FII)	CP043737.1	pN17EC0616-1
DSL_15	*qnr*S1	90	4316	-	-	-
DJF_13	*qnr*S1	67	4576	-	-	-

“-” indicated unknown.

## Data Availability

The original contributions presented in this study are included in the article/Appendix A. Further inquiries can be directed to the corresponding author.

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
