# Peer review of "Resistance Mechanisms of Fluoroquinolone in Escherichia coli Isolated from Taihe Black-Boned Silky Fowl Exhibiting Abnormally Slow Fluoroquinolone Metabolism in Jiangxi, China"

_antibiotics, 2025, doi:10.3390/antibiotics14090955_

Round 1
Reviewer 1 Report
Comments and Suggestions for Authors
Dear authors,
Please find attached my suggestions

Reviewer 2 Report
Comments and Suggestions for Authors
I have following comments for authors
-
- line 130: Author mentioned that they have identified other mutations however, details regarding those mutations were not provided
- Elaborate on the type of platform being used for sequencing and the tool used for creating genomic assemblies.
- It is recommend to provide reference, version and accession date with the tools used in the study
- In figure 2, author had shown association of other ARGs with FQ resistance and retention. However, it is unclear (i) if ARGs were identified in same vicinity as PMQR (ii) prevalence of other ARGs in isolates
- Elaborate on how sampling was performed. That should include season, how isolates were recovered.
- Author mentioned that FQ- resistant E. coli isolates were prevalent in TBSF . How they came up to this conclusion is not clear. the term prevalent is comparative in this context, implying a high occurrence relative to non TBSF. The study didn't include and investigate the prevalence in non TBSF farms. The study only did the characterization of FQ resistant isolates from TBSF farms.
- Please provide full form for each abbreviations in parentheses the first time it is used in the text.
Reviewer 3 Report
Comments and Suggestions for Authors
The manuscript “Resistance Mechanisms of Fluoroquinolone in Escherichia coli Isolated from Taihe Black-Boned Silky Fowl in Jiangxi, China” provides an important contribution to the field of antimicrobial resistance in poultry. The focus on Taihe Black-Boned Silky Fowl, with its slow fluoroquinolone metabolism, adds novelty and clear ecological relevance. The study design is robust, combining phenotypic testing with whole-genome sequencing to elucidate resistance mechanisms.
The results are highly informative. The predominance of QRDR mutations over PMQR determinants as the main resistance drivers is well demonstrated, and the discovery of the GyrA-S83Q mutation is both novel and significant. The network analysis linking fluoroquinolone residues with resistance phenotypes and genes offers a systems-level perspective that strengthens the conclusions.
The discussion is thorough and well contextualized, noting a possible evolutionary shift from plasmid-mediated to chromosomal resistance mechanisms. This observation is particularly thought-provoking and highlights the broader implications for resistance surveillance and control in poultry production systems.
Overall, this is a rigorous, original, and well-presented manuscript.
Author Response
Quality of English Language. The English could be improved to more clearly express the research.
We have revised the English language throughout the manuscript to enhance clarity and better convey the conclusions of this study.
Round 2
Reviewer 1 Report
Comments and Suggestions for Authors
Dear authors,
Thank you for revising the manuscript. In general the manuscript has been improve.
Please find below some few additional comments.
Line 336-337 : Revised by ‘The breast meat, feed, and egg samples were collected from the same batch as the fecal samples.
Line 336-337 : Please include in the text, the reasons why you have used feces and feed for residues analysis
Please clearly include the limitation of the study at the end of the discussion sessions
